# Anisotropy-mediated reentrant localization

Xiaolong Deng[1,2⋆], Alexander L. Burin[3] and Ivan M. Khaymovich[4,5,6]

**1** Leibniz-Rechenzentrum, Boltzmannstr. 1, D-85748 Garching bei München, Germany
**2** Institut für Theoretische Physik, Leibniz Universität Hannover,
Appelstr. 2, 30167 Hannover, Germany
**3** Department of Chemistry, Tulane University, New Orleans, Louisiana 70118, USA
**4** Max-Planck-Institut für Physik komplexer Systeme,
Nöthnitzer Straße 38, 01187-Dresden, Germany
**5** Institute for Physics of Microstructures, Russian Academy of Sciences,
603950 Nizhny Novgorod, GSP-105, Russia
**6** Nordita, Stockholm University and KTH Royal Institute of Technology,
Hannes Alfvéns väg 12, SE-106 91 Stockholm, Sweden

⋆ Xiaolong.Deng@itp.uni-hannover.de

## Abstract

We consider a 2d dipolar system, $d = 2$, with the generalized dipole-dipole interaction $\sim r^{-a}$, and the power $a$ controlled experimentally in trapped-ion or Rydberg-atom systems via their interaction with cavity modes. We focus on the dilute dipolar excitation case when the problem can be effectively considered as single-particle with the interaction providing long-range dipolar-like hopping. We show that the spatially homogeneous tilt $\beta$ of the dipoles giving rise to the anisotropic dipole exchange leads to the non-trivial reentrant localization beyond the locator expansion, $a < d$, unlike the models with random dipole orientation. The Anderson transitions are found to occur at the finite values of the tilt parameter $\beta = a$, $0 < a < d$, and $\beta = a/(a - d/2)$, $d/2 < a < d$, showing the robustness of the localization at small and large anisotropy values. Both exact analytical methods and extensive numerical calculations show power-law localized eigenstates in the bulk of the spectrum, obeying recently discovered duality $a \leftrightarrow 2d - a$ of their spatial decay rate, on the localized side of the transition, $a > a_{AT}$. This localization emerges due to the presence of the ergodic extended states at either spectral edge, which constitute a zero fraction of states in the thermodynamic limit, decaying though extremely slowly with the system size.



# 1  Introduction

With the realization of Anderson localization [1] of matter waves in optical lattice and of light [2], many extensions of disordered quantum systems are proposed [3] and implemented with and without interactions. A few of notable examples are vibrational modes of polar molecules [4], Rydberg atoms [5, 6], nitrogen vacancy centers in diamond [7], magnetic atoms [8, 9], photonic crystals [10], nuclear spins [11], trapped ions [12, 13] and Frenkel excitations [14].

In all these systems power-law decaying interactions are ubiquitous [3]. In addition, in the experiments of ultracold atoms the exponent $a$ of this power-law decay can be precisely controlled in a wide range, $0 < a < 2$ [12, 13] and $a = 3$ or $a = 6$ in [5, 6]. If the excitations in such systems are dilute, the long-range interaction induces the flips of far-away excitations. Thus, this problem has an effective single-particle description of (nearly) non-interacting excitations, where the above interacting term works as the power-law decaying excitation-flip hopping. Usually such excitations have internal degrees of freedom, similar to the dipole orientation. For homogeneous orientation of all such "dipoles", perpendicular to their plane, the corresponding disordered model has *deterministic isotropic* long-range hopping. Recent studies show that such models in the dimensionality $d = 1$ [15–17] and $d = 3$ [18], with fully-correlated hopping terms are localized even beyond the locator expansion convergence ($a < d$ for the power-law interaction). In particular, isotropic power-law hopping models $1/r^a$ show the power-law localization with the duality between the perturbative regime, $a > d$, and beyond it, $a < d$ [15, 16]. Note that for all $d \leq 2$ only the measure zero of the states located at one the spectral edges might be delocalized in such models.

As an experimentally feasible setup of the above generalized dipolar system, one can consider a set of ions, trapped in individual microtraps, which allows for arbitrary geometries and easy control over the effective anharmonicity of the spatial ion motion near the microtrap minima. Spin-dependent optical dipole forces, applied to such ionic crystal, create long-range

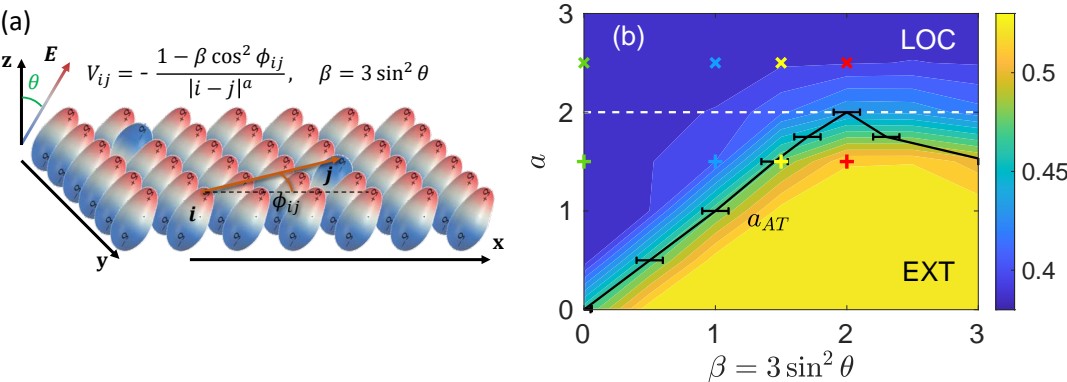

Figure 1: **Model and phase diagram.** (a) Two-dimensional (2d) lattice of quantum dipoles with dipole-dipole anisotropic interaction $(1 - \beta \cos^2 \phi_{ij})/|i-j|^a$, with few dipolar excitations (see dipoles down shown by blue tops). $i = (i_x, i_y)$ is the coordinate vector of $i$th dipole, $a > 0$ is the generalized power-law decay exponent. The anisotropy parameter $\beta = 3 \sin^2 \theta$ is governed by the homogeneous tilt angle $\theta$ of all dipoles by the electric field $E$, tilted from the normal $z$-axis towards the lattice plane. $\phi_{ij}$ is the angle between the spatial 2d vector $\mathbf{i} - \mathbf{j}$ and the $x$-axis-aligned projection of the electric field to the plane. (b) The phase diagram of the anisotropic 2d dipole model with dilute excitations and the on-site disorder. The color plot shows the $r$-statistics at $L = 200$, averaged over disorder $\langle r \rangle(E)$ and then over the spectral bulk $[-W/2, W/2]$, $\langle r \rangle$, the black solid line $a = a_{AT} = \min[\beta, \beta/(\beta - 1)]$ separates the localized ("LOC") phase, $a > a_{AT}$, from the extended phase, $a < a_{AT}$, while the black error bars correspond to these transition points, extracted from its finite-size scaling, Fig. 3. According to the analysis, the energy-resolved $r$-statistics $\langle r \rangle(E)$ is homogeneous across the spectral bulk away from the critical point (see Fig. 2). Thus, the finite-size analysis is done on the spectral-averaged $\langle r \rangle$ and the crossing point of the finite-size curves occurs at the critical value $\langle r \rangle \simeq 0.47$. The selected points with symbols "+" and "×" of the same colors (used in further figures) indicate the duality of power-law localization of wave functions for $a < d = 2$ and $a > d$.

effective spin-spin interactions and allow the simulation of spin Hamiltonians that possess nontrivial phases and dynamics (see, e.g., [12]). Tailoring the optical forces one can generate arbitrary interactions between spins. Our findings could be observed in the flip-flop spin-model, as well as in the phonon hopping model itself.

Another way to realize long-range anisotropic model would be to use the dipole radiation in a 2d photonic crystal near the Dirac cone (i.e., dipolar interaction mediated by the photonic Dirac cone between atoms), see Ref. [19] in which the authors obtain effective long-range interactions $1/r^{1/2}$, based on the results of Ref. [20]. The $1/r$ anisotropic hopping can be as well relevant for 2d polaritons [21]. One should notice that this interaction emerges at distances exceeding the wavelength. Consequently, the interaction amplitudes are complex numbers [19], and our consideration might need modifications there.

For dimensionality larger than $d > 1$, the above experimentally feasible dipolar-like systems are also characterized by common anisotropy which may have drastically different physics from the isotropic case. Usually the anisotropic terms are considered as quasi-disorder [22–25] and in the case of random and heterogeneous dipole orientations they lead to the localization-delocalization transition at $a = d$. In this paper we show that the situation is more subtle. In the case of homogeneous dipole anisotropy, relevant for the experiments in the electric

field, Fig. 1(a), this anisotropy gives rise to the *reentrant* localization phase diagram, Fig 1 (b) beyond the locator expansion, i.e. at $a < d$.

Thus, in order to combine measure zero of delocalized states from the isotropic case, $d \leq 2$, with the possibility of anisotropy, $d > 1$, we focus on a two-dimensional, $d = 2$, quantum dipolar system with the on-site disordered chemical potential and add a spatially homogeneous angular anisotropy of an effective dipolar form, Fig. 1(a). We show that the Anderson localization beyond the locator expansion is robust to the tilt, $\beta = 3\sin^2\theta$, homogeneous for all dipoles, up to a finite critical tilt value, Fig. 1(b), unlike the models with uncorrelated random off-diagonal hopping (see, e.g., [26,27]). Moreover, we demonstrate that the anisotropy leads to the reentrant character of localization showing localized eigenstates both at small (nearly isotropic) and large (strongly anisotropic) tilt. Such systems bridge the gap between models with deterministic and random interactions and bring new dimensions of anisotropy-mediated localization to the field of long-range systems. The extensive numerical simulations showing consistent behavior of level statistics and spatial wave-function properties are analytically supported by the renormalization group analysis (similar to [17,18]) and the newly developed matrix inversion trick [16].

## 2 Model and its symmetry

We consider the model describing dilute polar excitations propagating via dipole-flips (induced by their dipole-dipole interaction) on a square lattice of sites $\{i = (i_x, i_y)\}$ of size $L$, $i_x, i_y = 0, 1, \ldots, L-1$, Fig. 1(a), with the Hamiltonian

$$H = -\sum_{i,j} \frac{1 - \beta \cos^2\phi_{ij}}{r_{ij}^a} |i\rangle\langle j| + \sum_i \mu_i |i\rangle\langle i|, \tag{1}$$

where $\{|i\rangle\}$ are site basis states, $\mu_i \in [-\frac{W}{2}, \frac{W}{2}]$ is on-site disorder uniformly distributed over the above interval, the hopping term depends on the distance $r_{ij} = \sqrt{(i_x - j_x)^2 + (i_y - j_y)^2}$ between two lattice sites and its angle $\phi_{ij}$ with respect to the electric-field projection to the plane, i.e. $x$-axis. The effective single-particle hopping model (1) is obtained from the model of dipoles with dipole-dipole interactions as these interactions induce effective anisotropic transfer of excitations between sites via dipole-flips (see, e.g., [22,28]). The anisotropy parameter $\beta = 3\sin^2\theta$ is introduced by analogy to the experimental setup of dipolar molecules, Fig. 1(a), and is related to the homogeneous tilt angle $\theta$ of dipoles with respect to the $z$-axis. In this work we restrict our consideration to the physical values of $0 \leq \beta \leq 3$.

The isotropic limit, $\beta = 0$, considered in an early principle paper by Burin and Maksimov [18] for $a = d = 3$ and investigated in details for $d = 1$ in [15,16] represents a newly discovered universality class of long-range models with fully-correlated hopping. It is these complete correlations that bring destructive interference of long-range hops back into play, similarly to the standard weak and Anderson localization case, and localize the bulk of the system for all values of $a$ at $d \leq 2$. Here and further in the paper we mostly focus on the localization-delocalization transition in the spectral bulk, but not on the nonergodic wavefunction properties.

In the opposite limit of a long-range model with *fully uncorrelated random-sign* hopping $h_{ij}/r_{ij}^a$ [22–24,29,30] it is well-known that the localization occurs only for $a > d$, while the ergodic delocalization spans over the entire range $a < d$. The pure $d$-dimensional dipolar case of our model, $\beta = d$, (initially considered in [22–24,31] for different $d$) leads to the

same result, see Fig. 1(b).[1] Note that in general for such long-range models the disorder amplitude plays a subleading role, changing only the size of the wave-function "head" close to the maximal point, beyond which the wave-function decays polynomially.

One may naively expect that the intermediate case of $0 < \beta < d$ is similar to the perturbation of the fully-correlated model ($\beta = 0$) by a fraction $\epsilon \sim \beta/d$ of random-sign hopping $(1 + \epsilon h_{ij})/r_{ij}^a$ as finite $\beta$ works as a kind of quasi-disorder. However, in the latter model any $\epsilon > 0$ immediately delocalizes all the spectral states at $a < d$ as shown in [15, 26], which is not consistent with the phase diagram, obtained numerically and shown in Fig. 1(b).

Instead, in the anisotropic model (1) there is a *finite* tilt $\beta_{AT}(a)$ up to which the Anderson transition survives

$$\beta_{AT}(a) = a \quad \Leftrightarrow \quad a_{AT}(\beta) = \beta, \quad 0 \leq \beta_{AT}, a \leq 2. \tag{2}$$

This is the main result of the paper summarized in the phase diagram, Fig. 1(b), showing the localization properties of the bulk states, which is obtained from extensive numerical simulations.

Note that the Hamiltonian (1) obeys the $\pi/2$-rotational symmetry of a square lattice, $\phi_{ij} \leftrightarrow \phi_{ij} + \pi/2$, combined with the disorder strength $W$, the eigenenergy $E_n$, and the tilt $\beta$ rescaling

$$W \leftrightarrow \frac{W}{1-\beta}, \quad E_n \leftrightarrow \frac{E_n}{1-\beta}, \quad \beta \leftrightarrow \frac{\beta}{\beta - 1}, \tag{3}$$

because the hopping term goes under this transformation to

$$\frac{1 - \beta \cos^2\left(\phi_{ij} + \pi/2\right)}{r_{ij}^a} = \frac{1 - \beta \sin^2 \phi_{ij}}{r_{ij}^a} = (1 - \beta)\frac{1 - \frac{\beta}{\beta-1}\cos^2 \phi_{ij}}{r_{ij}^a}. \tag{4}$$

This symmetry relates the interval $0 < \beta < 2$ to the ones $\beta < 0$ and $\beta > 2$ and causes the reentrant character of the above phase diagram. In addition, for $\beta > 1$ the negative factor $1 - \beta$, which rescales the energy and the disorder, corresponds simply to mirroring of the spectrum and moving of extended ergodic states forming measure zero of all states from the bottom to the top of the spectrum. Thus, further without loss of generality further we restrict ourselves to $0 < \beta < 2$.

## 3 Overview of the numerical and analytical results

The eigenfunctions $\psi_n(i)$ and eigenenergies $E_n$ of the Hamiltonian, Eq. (1), are numerically calculated by exact diagonalization for 2d square samples of the linear size $L$ from 75 to 280 (i.e., from $\sim 5500$ to $\sim 80000$ matrix size) and for $10^2 - 10^3$ random realizations of the diagonal disorder. The ratio level statistics, Figs. 1(b) and 2(b),

$$\langle r \rangle (E) = \left\langle \min\left(r_{n,1}, \frac{1}{r_{n,1}}\right)\right\rangle, \quad r_{n,1} = \frac{E_n - E_{n-1}}{E_{n+1} - E_n}. \tag{5}$$

is calculated across the entire spectrum, with the disorder averaging and binning over energies. It shows the Poisson value $\langle r \rangle = 2\ln 2 - 1 \simeq 0.3863$ for *all* spectral bulk states in the localized phase, and $\langle r \rangle \approx 0.5307$ of Gaussian orthogonal ensemble (GOE) [32,33] at the spectral edge

---

[1]However, in a recent paper of two of authors [35] it has been shown that the critical point $a = \beta = d = 2$ is more subtle: it shows diffusive transport and non-ergodic eigenstates like in [31] for strong diagonal disorder and superdiffusion with ergodic wave functions at small disorder.

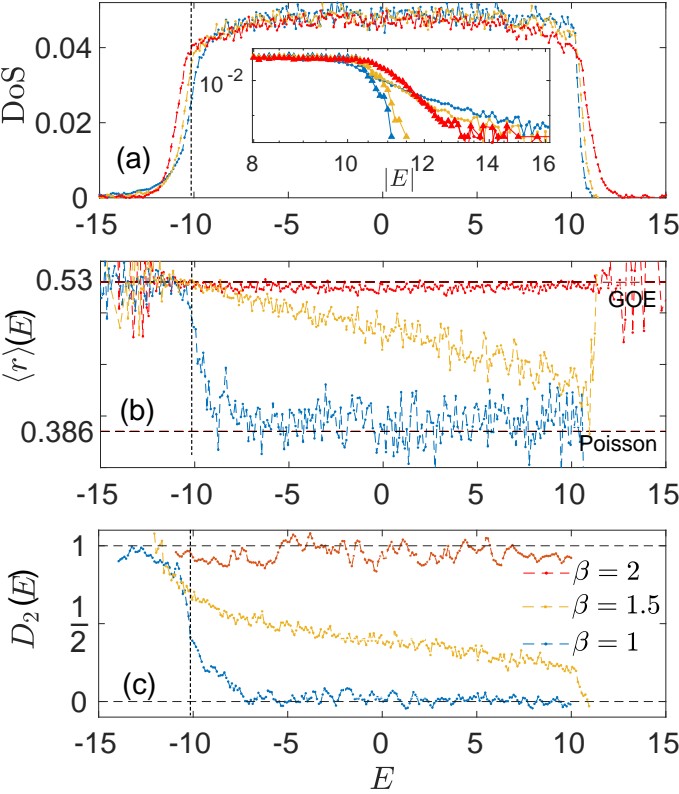

Figure 2: **Emergence of the finite-size mobility edge across the Anderson transition.** (a) global density of states (DOS), (b) energy-resolved level-spacing ratio $r$-statistics $\langle r \rangle(E)$, and (c) energy-resolved fractal dimensions $D_2$ for each eigenstate versus energy $E$ in the localized ($a = 1.5$, $\beta = 1$, blue), critical ($a = \beta = 1.5$, yellow), and delocalized ($a = 1.5$, $\beta = 2$, red) phases. Both panels (b) and (c) show localized (blue), critical (yellow) and ergodic (red) eigenstate properties in the spectral bulk. The bulk states are within the range $[-W/2, W/2]$, where we take $W = 20$. The inset to panel (a) shows power-law tails of DOS at either ($a > a_{AT}$) or both ($a < a_{AT}$) spectral edges beyond the bulk, which correspond to the ergodic states in panels (b) and (c). For the energy-resolved data the bins of the 50 adjacent states are used. The data for $D_2$ are extrapolated from $L = 100, 150, 200,$ and 250 with the corresponding number of disorder realizations 1000, 500, 100, and 50, respectively, see Sec. 4.2.1 for details. For the rest of the data $L = 250$. The vertical dashed lines provide the position of the finite-size mobility edge extracted from the finite-size data for $r$-statistics.

and for all eigenstates in the extended phase. The spectral-resolved fractal dimension $D_2(E)$, extracted as an exponent from the inverse participation ratio (IPR)

$$I_2(E_n) = \sum_i |\psi_n(i)|^4 \propto L^{-d \cdot D_2(E_n)}, \qquad (6)$$

shows consistent behavior in the localized ($D_2 \to 0$), critical ($0 < D_2 < 1$), and extended phases ($D_2 \to 1$), Fig. 2(c), for the spectral bulk states.[2] The non-trivial bulk-energy-dependence of $\langle r \rangle(E)$ and $D_2(E)$ at the critical point might be a result of the finite-size effect. As in this work we focus on the localized and ergodic phases and only on the location of the phase

---

[2]Note the non-standard definition of the fractal dimension: the dimension of the corresponding fractal in $d = 2$-dimensional space is given by $d \cdot D_2$.

transition, further we do not consider the above energy dependencies and average the data over the bulk of the spectrum, if not stated otherwise.

Basing on the above spectral-resolved data for many $(a, \beta)$ points, in the next sections we perform more deep analysis to determine the phase diagram of the bulk spectral states, shown in Fig. 1(b), as well as the fraction of ergodic spectral-edge states in the localized phase.

In Sec. 4.1 in order to determine the phase diagram, Fig. 1(b), and confirm the analytical prediction $\beta = \beta_{AT}(a)$, Eq. (2), the more detailed analysis of the finite-size scaling (FSS) of $r$-statistics has been performed.

In Sec. 4.2 in order to support the claim of the paper on anisotropy-mediated reentrant localization beyond the convergence of the locator expansion, $a < d = 2$ and confirm the consistency of the spectral and spatial measures, we provide the numerical analysis of the spectrum of fractal dimensions and of the eigenstate spatial decay from the maximum, inevitably showing the power-law localization of the spectral bulk states at all $a_{AT}(\beta) < a < d$ and ergodic delocalization otherwise. However, in the latter case the finite-size data convergence of the fractal dimension to 1 very slow, see Sec. 4.2.1, and the higher-order extrapolation in $1/\ln L$ or the linear one with irrelevant exponent [34] shows significant fluctuations, see [35] and Appendix A for details. The static data analysis of Sec. 4.2 is supported by the dynamics of the wave packet in Sec. 4.3, confirming the localization at $a_{AT}(\beta) < a < d$.

In Sec. 4.4 in order to understand the contribution of the spectral edge states, in the localized phase we determine the location of the finite-size mobility edge via the threshold in the $r$-statistics, see Figs. 2 and 9, as well as the fraction of the ergodic $r$-values. In addition we extract the fraction of ergodic IPR values from the energy-resolved data, sorted in increasing IPR value, Fig. 10. The consistency of the data and its agreement with the analytical predictions allows us to claim that the ergodic spectral-edge states form a measure zero of all the states in the thermodynamic limit, however, at finite sizes this fraction decay only as a power of the logarithm of the system size $L$.

Section 5 represents the corresponding analytical analysis: we show that it is the spectrum of hopping for the considered 2d dipolar model, Eq. (1), which provides the analytical prediction for the phase diagram in Fig. 1(b). The above idea is formalized by the renormalization group analysis, briefly given in Sec. 5.1, where the importance of the sign-constant spectrum of hopping is discussed. Section 5.2 is devoted to the analytical prediction of the fraction of ergodic eigenstates and based on the Ioffe-Regel criterion.

Section 6 concludes our consideration.

# 4 Numerical results

## 4.1 Finite-size flow of the ratio $r$-statistics

Taking into account the fact that $r$-statistics, Eq. (5), is homogeneous for *all* spectral bulk states in both ergodic and localized phase, we perform the finite-size scaling of the data $\langle r \rangle$, averaged both over disorder and the bulk of the spectrum. We determine the transition line $\beta = \beta_{AT}(a)$, Eq. (2) and the error bars in Fig. 1(b) via the change of finite-size flow of $\langle r \rangle(a, \beta, L)$ versus $L$, Fig. 3(a-d).

First, we describe the procedure of the finite-size collapse of the ratio $r$-statistics. For each value of the bare hopping decay rate $a$ the disorder- and spectral-average ratio $r$-statistics has been calculated for the range of anisotropy parameters $\beta$ and system sizes $L$ (see Fig. 3(a-d) for $a = 0.5$, $a = 1.0$, $a = 1.5$, and $a = 1.75$, respectively).

The first approximation of the transition $\beta = \beta_{AT}(a)$ is given by the crossing point of finite-size $r(\beta, L)$ curves, see Figs. 3(a-d). The intersection points of $r$-statistics versus $\beta$ for different

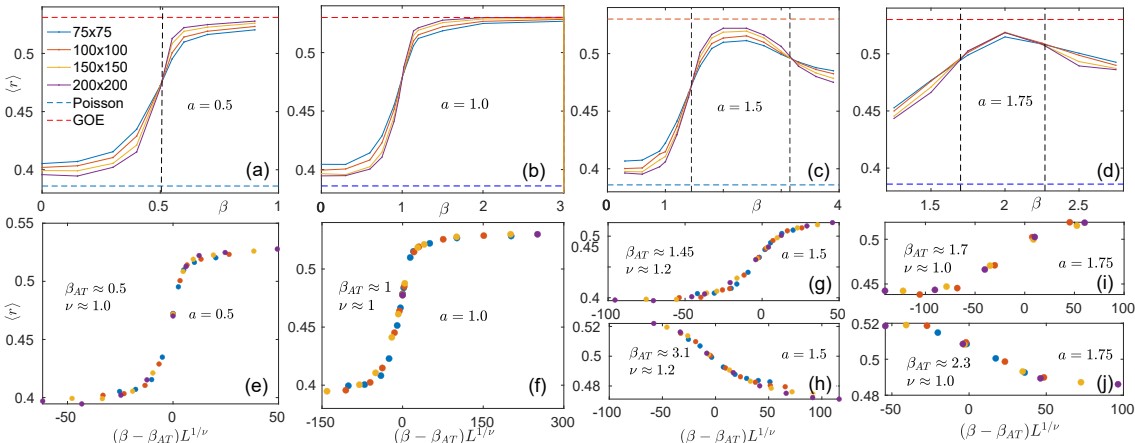

Figure 3: **Spectral-averaged ratio $r$-statistics versus the anisotropy parameter $\beta$.**
Level statistics, averaged over the spectral bulk $E \in [-W/2, W/2]$, versus anisotropy
$\beta$ at (a) $a = 0.5$, (b) $a = 1.0$, (c) $a = 1.5$, and (d) $a = 1.75$ for the disorder strength
$W = 20$ and different system sizes $L = 75$, 100, 150, and 200 (shown in legend)
with the corresponding number of disorder realizations 2000, 2000, 1000, and 400,
respectively. Dashed horizontal lines show the limiting ergodic ($r \simeq 0.53$) and Pois-
son ($r \simeq 0.386$) values. The vertical lines show the extracted anisotropy parameter
$\beta_{AT}$ at the localization transition. Panels (e-j) show the finite-size collapse of all 6
crossing points using $\langle r \rangle = R\left[(\beta - \beta_{AT})L^{1/\nu}\right]$, with critical values $\beta_{AT}$ and critical
exponents $\nu$.

system sizes correspond to the Anderson localization transition and agree quite well with the
analytical values shown in Fig. 1(b).

More accurate single-parameter collapse of all curves of the form

$$\langle r \rangle (\beta, L) = R(|\beta - \beta_{AT}|L^{1/\nu}). \tag{7}$$

provides best parameters $\beta_{AT}$ and $\nu$, see Fig. 3(e-j). It gives $\beta_{AT} = a \pm 0.05$ for $0 < \beta < 2$
and $\nu = 1.0 \pm 0.2$ for all considered $a$. At the transition line the mean $r$-statistics takes the
universal value $\langle r \rangle \approx 0.47$ independent of $a$. Note that the pairs of crossings $\beta_{AT}$ in Fig. 3(c, d)
are related to each other with respect to the symmetry (3) within the above mentioned error
bar, while the critical exponents are just the same. The black solid line in Fig. 1(b) shows the
result for the critical value of $\beta_{AT}$ extracted from Fig. 3, which coincides with the analytical
prediction, Eqs. (2), (3), within the $\sim 10$ %-errorbar.

## 4.2 Eigenstate properties: multifractal analysis and wave-function spatial decay

In addition to the spectral properties, we consider eigenstate ones. We focus on the multifractal
analysis, based on the spectrum of fractal dimensions $f(\alpha)$, fractal dimension $D_2$, as well as
on the spatial decay of the wave functions with the distance from their maxima.

### 4.2.1 Spectrum of fractal dimensions $f(\alpha)$ and fractal dimensions $D_q$

In this subsection we focus on the spectral-bulk properties, define the spectrum of fractal di-
mensions and provide the standard extrapolation procedure for it (see, e.g., [15,16,26,36,37])
and for the fractal dimensions $D_q$ [30], Eq. (6).

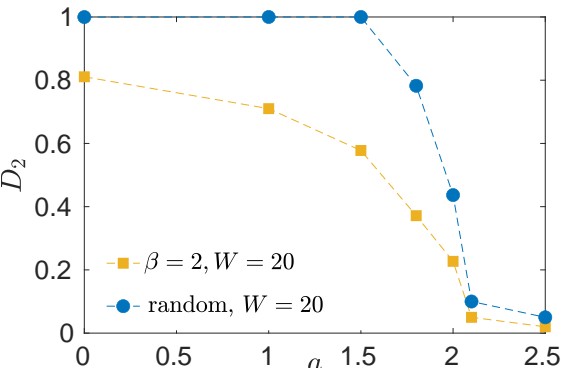

Figure 4: **Comparison of extrapolated fractal dimension $D_2$ versus power $a$** for the anisotropic model with fixed bare disorder $W = 20$ (yellow squares) and for the $2d$ power-law random banded model (blue circles). The anisotropy is taken to be $\beta = 2$. $D_2$ are extrapolated from $L = 100, 150, 200,$ and $250$ with the corresponding number of disorder realizations 1000, 500, 100, and 50, respectively.

The standard multifractal analysis is based on the generalization of the IPR (6) to the other wave-function moments:

$$I_q = \left\langle \sum_i |\psi_n(i)|^{2q} \right\rangle = c_q L^{d(1-q)D_q} \, , \tag{8}$$

where the scaling exponents $D_q$ are called fractal dimensions of the order $q$.

The finite-size fractal dimension is defined by the formula $D_q(L) = \ln I_q / (1-q) \ln L^d$ and the main contributions to it are given by the scaling exponent $D_q$ and the prefactor $c_q$ of IPR similarly to (12)

$$D_q(L) = D_q + \frac{(1-q)^{-1} \ln c_q}{\ln L^d} \, . \tag{9}$$

The resulting extrapolated $D_2$ is shown in Fig. 4 versus $a$ for $\beta = 2$. One can see there (yellow squares) the transition from localized phase $a > 2$ with $D_2 \to 0$ to the extended one, $D_2 > 0$, at $a < 2$. As a reference point (blue circles) we consider the generalization of the power-law random banded matrix (PLRBM) model [30, 38] to 2d by replacing the correlated factor $1 - \beta \cos^2 \phi_{ij}$ in (1) by a i.i.d. random numbers, and show the fractal dimension extrapolated using the simple linear formula (9). The latter confirm the known results $D_2 = 1$ for $a < d = 2$ and $D_2 = 0$ for $a > d$ with good accuracy. The discrepancy between these models in the extended phase is due to severe finite-size effects in anisotropic model (we address this issue in the Appendix A).

Next we consider the dual measure, namely the spectrum of fractal dimensions $f(\alpha)$, characterizing the multifractality of the states, which is defined via the probability distribution

$$\mathcal{P}(\ln |\psi_n(i)|^2) \sim L^{d(f(\alpha)-1)} \, , \tag{10}$$

of the logarithm of the wave-function intensity $\alpha = -\ln |\psi_n(i)|^2 / \ln L^d$ [30] and can be extracted directly from the histogram over $\alpha$ [16, 26, 36, 37, 39].

The relation between the spectrum of fractal dimensions $f(\alpha)$ and the fractal dimensions $D_q$ is given by the saddle-point approximation for the disorder averaged generalized IPR [30]

$$\langle I_q \rangle = L^d \int P(\alpha) L^{-dq\alpha} d\alpha = L^{d \max_\alpha (f(\alpha) - q\alpha)} \equiv L^{d(1-q)D_q} \iff D_q = \frac{\min_\alpha (q\alpha - f(\alpha))}{q-1} \, , \tag{11}$$

where under the disorder average the sum over coordinates is replaced by the factor $L^d$ and each summand is averaged over $P(\alpha)$. This confirms that $f(\alpha)$ contains all the information about $D_q$ via the above Legendre transform.

In addition, usually for the non-ergodic extended states in most cases $f(\alpha)$ obeys a so-called Mirlin-Fyodorov symmetry $f(1 - \delta\alpha) = f(1 + \delta\alpha) - \delta\alpha$ [30]. The ergodic extended state corresponds to a $\delta$-function at $\alpha = 1$,[3] while the localized state has $f(0) = 0$ and a certain (usually linear) form of $f(\alpha > 0) = k\alpha$, with $k = 0$ for exponential and $k > 0$ for power-law localization.[4]

As for the finite-size effects, due to the normalization condition of the probability distribution (10), one should take into account finite-size prefactors of $P(\alpha)$ which depend on the system volume $L^d$ slower than any power. In order to take this into account, we use the following expression $f(\alpha, L)$ at the finite system size $N = L^d$, $d = 2$

$$f(\alpha, L) = f(\alpha) + \frac{c_\alpha^{(1)}}{\ln L^d} + \frac{c_\alpha^{(2)}}{(\ln L^d)^2} + \dots , \tag{12}$$

with a certain $\alpha$-dependent constants $c_\alpha^{(k)}$, depending on the $P(\alpha)$-prefactors. Here and further we stick to the quadratic in $1/\ln L^d$ behavior in order to have reliable extrapolation.

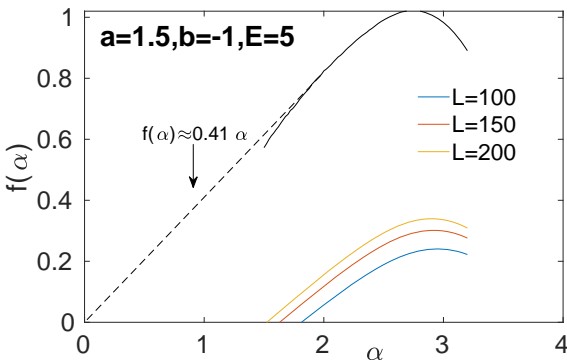

Figure 5: **Finite-size extrapolation of the multifractal spectrum** $f(\alpha)$ for the energy $E = 5$, disorder strength $W = 10$, $a = 1.5$, and $\beta = -1$. $f(\alpha)$ is extrapolated from $L = 100$, 150, and 200 with the corresponding number of disorder realizations 1000, 500, and 100, respectively.

The corresponding finite-size $f(\alpha, L)$ and extrapolated $f(\alpha)$ curves are given in Fig. 5 for a certain mid-spectrum energy $E = 5$ in the localized phase, $a = 1.5$, $\beta = -1$ and obey the normalization condition, $\max_\alpha f(\alpha) = f(\alpha_0) = 1$, of the probability distribution $\mathcal{P}(\alpha)$.

Generally, the position of the maximum $\alpha_0$ of $f(\alpha)$ and its slope $k = 1/\alpha_0$ corresponds to the effective power-law spatial decay of the wave function with the distance $r = |i - i_0|$ from its maximum $i = i_0$. Indeed, with the distance the eigenstate decays as $L^{-d\alpha} = |\psi_n(i)|^2 \sim r^{-\gamma(a)}$, $\gamma(a) = 2\max(a, 2d - a)$, while the number of states increases as the volume $L^{df(\alpha)} \sim r^d$. Thus, resolving these expressions with respect to $r$ one obtains

$$f(\alpha) = \frac{\alpha}{\alpha_0}, \quad \alpha_0 = \frac{\gamma(a)}{d} = \max(a, 2d - a), \tag{13}$$

which is confirmed by the numerical simulations, Fig. 5.

---

[3]With the additional tail $f(\alpha) = (3 - \alpha)/2$ for $\alpha > 1$ due to the statistics of de Broglie-like oscillations of $\psi_n$, see, e.g., the Supplemental Information in [36] and [39] for details.

[4]Note that $k = 1/2$ corresponds to the critical localization as $f(\alpha) = \alpha/2$ in this case still obeys the Mirlin-Fyodorov symmetry [30].

In the model (1) at $a = 1.5 < d = 2$ as an example (solid lines) the spectrum of fractal dimensions of the bulk spectral states, Fig. 6(a), shows power-law localized ($\beta = 1$, blue), critical ($\beta = 1.5$, yellow), and ergodic ($\beta = 2$, red) behavior in the localized phase, at the transition, and in the extended phase, respectively. The corresponding data at $a = 2.5 > d$ is always power-law localized (dashed lines) with $\gamma(a) = a$.

At the critical $a = d = 2$ of the convergence of the locator expansion the wave-function behavior is consistent with the critical localization, Fig. 7, $f(\alpha) \simeq k\alpha$, with $k = 1/2$ corresponding to the critically localized eigenstate and the spatial decay [17].

The wave-function spatial decay $\langle \ln |\psi_n(i)|^2 \rangle$ of the typical wave-function intensity $|\psi_n(i)|^2$ with the distance $r = |i - i_0|$ from its maximum $i = i_0$ suggested as the localization measure in [15] and used in [16, 17, 26] shows the consistent power-law localization with the decay rate being dual with respect to the critical line $\alpha = d = 2$

$$|\psi_n(i)| \sim r^{-a} \quad \text{for } a > d, \tag{14a}$$

$$|\psi_n(i)| \sim r^{a-2d} \quad \text{for } a < d, \tag{14b}$$

as in [15–17] in the whole range of anisotropy parameter $\beta$ in the localized phase, $\alpha > \alpha_{AT}(\beta)$, Eq. (2), see blue lines in Fig. 6(b) and in Appendix B.

At the self-dual line $a = d = 2$ of (14), Fig. 7, the wave-function behavior is consistent with the above critical localization $f(\alpha) \simeq \alpha/2$ and corresponds to the localized eigenstate and the spatial decay [17]

$$|\psi_n(i)| \sim r^{-d} (\ln r)^{-2} . \tag{15}$$

The pure 2d dipole point $a = \beta = 2$ considered in [31] and revisited in [35] is exempted here as it shows the transition from ergodicity to localization over the disorder amplitude.

Both (14) and (15) can be understood in terms of the renormalization group (RG) analysis similar to the one in [17, 18], given in Sec. 5.1.

## 4.3 Wavepacket dynamics. Return probability

In addition to the static properties given in the previous section by multifractal analysis and wave-function spatial decay, we confirm the eigenstate localization dynamically.

For this purpose we initialize the wave packet with the delta function at time $t = 0$,

$$\psi(0) = \delta(x - x_0) = \sum_n \psi_n^*(x_0, 0)\psi_n(x, 0), \tag{16}$$

and compute the evolution of it in time

$$\psi(t) = \sum_n \psi_n^*(x_0, 0)\psi_n(x, t) = \sum_n \psi_n^*(x_0, 0)\psi_n(x, 0)e^{-iE_n t}, \tag{17}$$

by considering the survival probability defined as [40–42]

$$R(t) = |\langle \psi(0)|\psi(t)\rangle|^2 = \sum_{n,m} |\psi_n(x_0, 0)|^2 |\psi_m(x_0, 0)|^2 e^{-i(E_n - E_m)t} . \tag{18}$$

which is an important dynamical measure relevant also for many-body localization [43, 44].

By definition at time $t = 0$ the survival probability equals unity and then as time evolves it decays (with some revivals) to the constant value at long times

$$R(t \to \infty) = \sum_n |\psi_n(x_0, 0)|^4 , \tag{19}$$

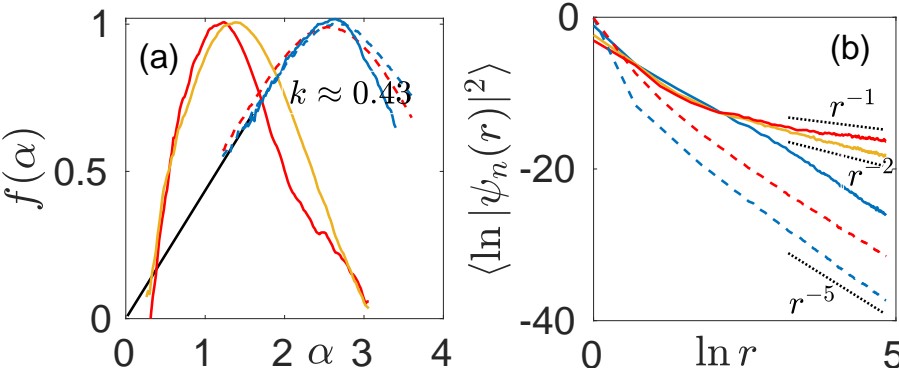

Figure 6: **Spatial properties in the spectral bulk.** (a) spectrum of fractal dimensions $f(\alpha)$ and (b) power-law spatial decay of eigenstates in the bulk of the spectrum for $a = 1.5$ (solid), 2.5 (dashed), and $\beta = 1$ (blue), 1.5 (yellow), 2 (red). The panel (b) confirms the duality of power-law spatial decay rate $\gamma(a) \approx \gamma(2d - a)$ [15–17] in the localized phase ($\beta < a < d$ or $a > d$), also supported by the slope $k < 0.5$ of $f(\alpha)$ in panel (a). $f(\alpha)$ is extrapolated from $L = 75$, 100, 125, 150, 200, 225 and 250 with the corresponding number of disorder realizations from 2000 for $L \leq 125$ to 300 for $L \geq 225$, see Sec. 4.2.1 for details. and with the disorder amplitude $W = 20$. For the spatial decay $L = 250$ and $W = 20$ for $a = 1.5$, for $a = 2.5$ we choose bigger $W = 200$ in order to make the power-law tail dominant on moderate sizes.

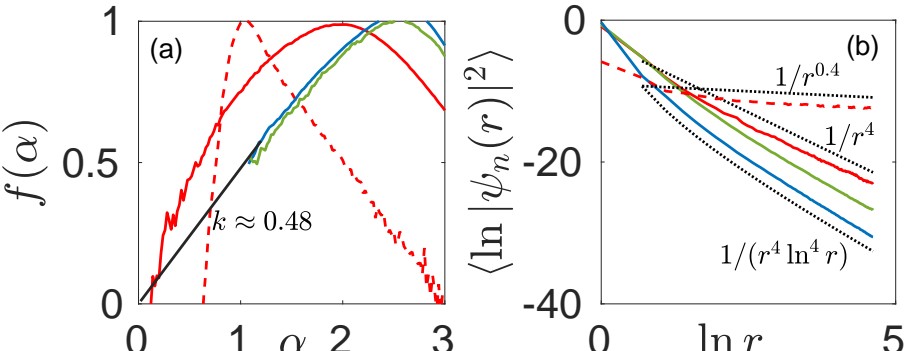

Figure 7: **Spatial properties at the dual line $a = d = 2$.** (a) spectrum of fractal dimensions $f(\alpha)$ and (b) power-law spatial decay of eigenstates in the bulk of the spectrum at the self-dual line $a = 2$ of (14) for $\beta = 1$ (blue), 2 (red), 3 (green). The linear behavior of $f(\alpha)$ with the slope close to $k = 0.5$ supports the critical localization for $\beta \neq 2$. The exceptional point $a = \beta = 2$ shows the transition from ergodicity ($W = 4$, dashed) to localization ($W = 40$, solid) over the disorder amplitude. The disorder strength for $\beta = 1$, 3 is $W = 40$. $f(\alpha)$ is extrapolated from $L = 75$, 100, 125, 150, 200, 225 and 250 with the corresponding number of disorder realizations from 2000 for $L \leq 125$ to 100 for $L = 250$, see Sec. 4.2.1 for details. For the spatial decay $L = 200$.

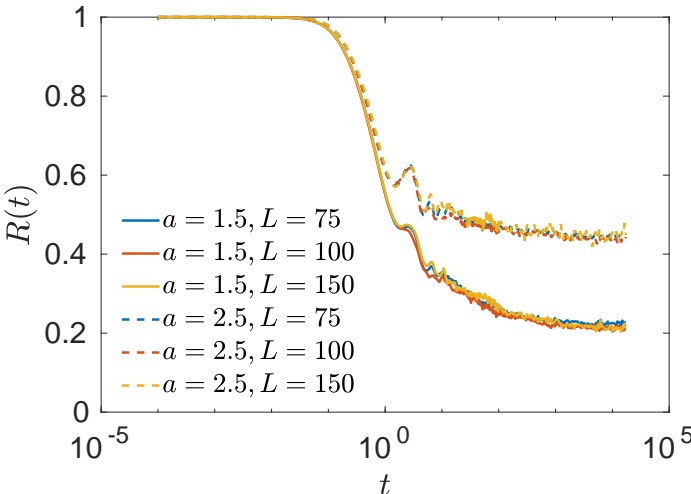

Figure 8: **Survival probability of the delta-peak initialized wave packet in the localized phase** for the disorder strength $W = 20$, $\beta = 0.3$, and $a = 1.5$ (solid lines), $a = 2.5$ (dashed lines) for different system sizes $L = 75$, 100, and 150 with the corresponding number of disorder realizations 1000, 750, and 400, respectively.

analogously to the IPR with the summation over energies, but not coordinates. The scaling of the latter measure with the system size $L^d$ shows the localization properties of the wave packet and, thus, of the underlying eigenstates.

Figure 8 shows the survival probability at $\beta = 0.3$, $a = 1.5$ and $a = 2.5$, corresponding to the localized phase $\beta < a < d = 2$ and $a > d$, respectively, for several system sizes, averaged over the disorder realizations and several initial coordinated $x_0$. From the data it is clearly seen that in both cases the limiting value (19) does not scale with the system size confirming the localization of the eigenstates. The larger limiting value of $R(t)$ for $a > d$ corresponds to the smaller localization region of localized states with respect to $\beta < a < d$ according to the renormalization group predictions.

## 4.4 Finite-size mobility edge and the fraction of ergodic states

From the spectral-resolved measures, considered in Sec. 3, one can extract the position of the finite-size mobility edge $-E^*$ below which all the states are ergodic both at $a < a_{AT}$ and $a > a_{AT}$, while the other ones are power-law localized above $E > -E^*$ for $a > a_{AT}$ and extended with smaller extrapolated $D_2$ for $a < a_{AT}$.

In order to determine $E^*$, first, we consider a threshold analysis of the energy-resolved $r$-statistics, see Fig. 9. The data on the right panel shows that the corresponding fraction of ergodic states $f_{\text{erg}} = \int_{E<-E^*} \rho(E)dE/L^d$ in the localized phase decays with the system size $L$, but does it logarithmically slowly, according to the Ioffe-Regel criterion, considered in Sec. 4.4.

Next, we focus on the estimation of the fraction of ergodic high-energy states in the localized state at $0 < \beta < a < 2$ from the IPR. For this we consider the plot of energy-dependent IPR values sorted in increasing order for different system sizes versus the renormalized fraction of the states $(n/L^d)^{3-a}/\ln L$, see Fig. 10. Panels (a) and (b) show the IPR itself $I_2$ and its renormalization $L^d \cdot I_2$ in order to confirm the scaling of the localized and ergodic states, respectively, given by the right panel of Fig. 9 for the ratio $r$-statistics versus energy.

The consistency of the above spectral and spatial data leads us to the conclusion that in the localized phase, $a > a_{AT}$, there is measure zero of the delocalized edge states which can be neglected in the thermodynamic limit.

In the next section we provide an analytical consideration of the model (1) and explain the non-trivial bulk-spectrum phase diagram with the localization beyond the convergence of the locator expansion and its relation to the decaying fraction $f_{erg}$ of ergodic high-energy states.

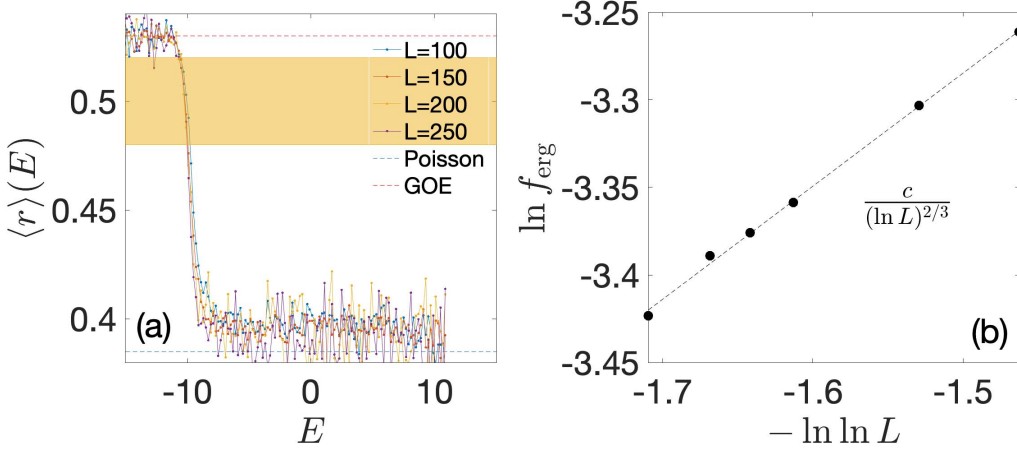

Figure 9: **Threshold analysis of ratio $r$-statistics versus the energy $E$.** (left) Energy-resolved $\langle r \rangle(E)$ for the disorder strength $W = 10$, $a = 1.5$, and $\beta = 0.3$ for different system sizes $L = 100$, $150$, $200$, and $250$ with the corresponding number of disorder realizations 1000, 1000, 150, and 50, respectively. Dashed horizontal lines show the limiting ergodic ($r \simeq 0.53$) and Poisson ($r \simeq 0.386$) values. The orange shaded region shows the threshold $r \in [0.48, 0.52]$ used for the extraction of the fraction $f_{erg}$ of the ergodic states in the main text. (right) Fraction $f_{erg}$ of ergodic extended states below the finite-size mobility edge, $E < -E^* < 0$, extracted from the data in the left panel in the localized phase, $a = 1.5$, $\beta = 0.3$, versus the system size $L$. The number of disorder realizations is from 2000 for $L \leq 100$ to 150 for $L = 250$. Numerical $f_{erg}$ (symbols) is consistent with analytical predictions (dashed line) in (23).

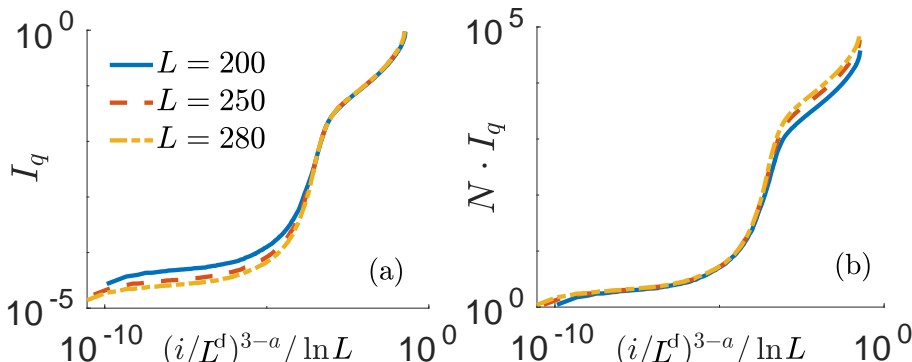

Figure 10: **Energy-resolved inverse participation ratio sorted in increasing order versus renormalized state index** (a) IPR itself showing collapse at the localized states and (b) IPR renormalized to the system size $N = L^d$ showing the collapse for ergodic states. The disorder strength for $a = 1$, $\beta = 0.3$ is $W = 20$. Finite size data is represented for $L = 200$ (solid blue), 250 (dashed red), and 280 (dash-dotted yellow) with the corresponding number of disorder realizations 100, 80, and 50, respectively.

# 5 Analytical methods and results

The non-trivial phase diagram for the bulk spectrum, Fig. 1(b), and anisotropy-mediated reentrant localization can be understood from the structure of the high-energy states and spectrum of hopping in both isotropic, $\beta = 0$, and anisotropic, $\beta > 0$ cases of a dipole system (1). Indeed, the hopping term $\sum_{\mathbf{i,j}} |\mathbf{i}\rangle\langle\mathbf{j}|(1 - \beta\cos^2\phi_{ij})/|i-j|^a$ in (1), which is translation invariant and therefore can be diagonalized in the momentum basis $|\mathbf{q}\rangle = \sum_n e^{i\mathbf{qn}}|\mathbf{n}\rangle/L^{d/2}$ as $\sum_{\mathbf{q}} V_{\mathbf{q}}|\mathbf{q}\rangle\langle\mathbf{q}|$, diverges at small $|\mathbf{q}| < q_* \ll 1$ and $a < d = 2$

$$V_{\mathbf{q}} = -\int_0^\infty r\,\mathrm{d}r \int_0^{2\pi} \mathrm{d}\phi\, e^{iqr\cos(\phi-\phi_q)} \frac{1 - \beta\cos^2\phi}{r^a} = c_a q^{a-2}\left[\beta - a - (2-a)\beta\cos^2\phi_{\mathbf{q}}\right]. \quad (20)$$

Here $c_a = -\frac{\pi\Gamma(-a/2)}{2^a\Gamma(a/2)} > 0$, $\Gamma(a)$ is a Gamma-function, and the momentum $\mathbf{q} = \frac{\pi}{L}(m_x, m_y) = q(\cos\phi_{\mathbf{q}}, \sin\phi_{\mathbf{q}})$ is written in polar coordinates $q$, $\phi_{\mathbf{q}}$, with $m_x, m_y = 0, 1, \ldots, L-1$, see Appendix C for the calculation details.

As a result of this divergence, there are eigenstates of the Hamiltonian (1) at high energies $|E_{\mathbf{q}}| \simeq |V_{\mathbf{q}}| > E^*$, with $|E_{\mathbf{q}}| \gg W$ and $|q| < q_*$, which are barely affected by the on-site disorder and represented by superpositions of plane waves only with small momenta $|q| < q_*$. Note that for the isotropic case $\beta = 0$ these divergent energies appear only at the negative side of the spectrum (see Fig. 2). As the corresponding momenta $|q| < q_*$ in $d < 3$ constitute an extensive number, but zero fraction of all $q$, these states are non-ergodic or even localized in the momentum space, meaning that they should be diffusive or ballistic in the coordinate basis.

Although the above *exact* eigenstates with large energies $|E_{\mathbf{q}}| \simeq |V_{\mathbf{q}}| > E^*$ constitute a *zero* fraction of all states, they give the dominant contribution to the hopping term due to their eigenvalues

$$\sum_{\mathbf{q}} V_{\mathbf{q}}|\mathbf{q}\rangle\langle\mathbf{q}| = \sum_{|E_{\mathbf{q}}|>E^*} E_{\mathbf{q}}|E_{\mathbf{q}}\rangle\langle E_{\mathbf{q}}| + J_{\mathrm{res}}. \quad (21)$$

Here we keep the index $\mathbf{q}$ for these states as their large energies are almost insensitive to the disorder term and therefore close to their bare kinetic values $V_{\mathbf{q}}$. The last term $J_{\mathrm{res}}$ in r.h.s. contains the summation over the small energies $|E_{\mathbf{q}}| < E^*$ and perturbative deviations between $|\mathbf{q}\rangle$ and $|E_{\mathbf{q}}\rangle$.

The action of the total kinetic term on the bulk eigenstates $E_n > E^*$, being orthogonal to the above ones, $\langle E_{\mathbf{q}}|E_n\rangle = 0$, is equivalent to the action of the residual hopping term $J_{\mathrm{res}}$ only. If there is no compensation in the first sum of Eq. (21), the residual term $J_{res}$ should be much smaller than the total kinetic term, having substantially faster spatial decay. This effectively short-range hopping $J_{res}$ leads to the localization of the entire spectral bulk providing a new localization mechanism due to the presence of measure zero of delocalized high-energy states orthogonal to them.[5]

These simple arguments work provided the extended high-energy states appear on the *only* spectral edge and, thus, their contribution to (21) is *not* compensated by the states from the opposite one. The effect of extended spectral edge states has been partially understood for the case of the only such state in terms of cooperative shielding in [46] and explained in details for the general case $a \geq 0$, $d = 1$ by the matrix inversion trick in [16, 26] and by the renormalization group in [17]. In our model (1), the condition that $V_{\mathbf{q}}$ diverges at small $|q| < q_*$, $a < d$, and do not change the sign for different momentum orientations $\phi_{\mathbf{q}}$ in order to have high-energy states on the *only* spectral edge is given by

$$V_{\mathbf{q}}(\phi_{\mathbf{q}})/V_{\mathbf{q}}(0) > 0 \quad \Leftrightarrow \quad a|\beta - 2| > |a - 2|\beta. \quad (22)$$

---

[5]Similar effects have been recently observed in non-integrable many body systems where the special spectral-edge states lead to the departure from the eigenstate thermalization hypothesis in the spectral bulk [45].



Figure 11: **Illustration of the spectral divergence and the localization at $a < d$** for (a) the isotropic, $\beta = 0$, (b) anisotropic localized, $0 < \beta < \beta_{AT}(a)$, and (c) anisotropic delocalized, $\beta > \beta_{AT}(a)$, cases. The top panel shows the one- or two-sided divergence of the spectrum, while the bottom illustration shows the tilt of the 2d dipolar system in the strong electric field.

It immediately provides the phase boundary of the localization $\beta < \beta_{AT}(a)$, valid for all $a$ and $\beta$, see Eq. (2) for $\beta < 2$ and the symmetry (3) for the rest values. In order to illustrate it, in Fig. 11 we show the kinetic spectrum $V_{\mathbf{p}}$ (top) and the corresponding 2d dipolar system in a tilted electric field for $a = 1 < d = 2$ for (a) isotropic, $\beta = 0$, and (b) anisotropic, $\beta = 0.75$ localized cases, as well as for (c) the delocalized one, $\beta = 2.25$. One can see that for the localized cases the spectrum diverges to the only direction, even at finite anisotropy, while its two-sided divergence immediately leads to the delocalization.

Note also that the power-law growth of the spectral-edge energies $E_{\mathbf{q}} \simeq V_{\mathbf{q}} \sim q^{a-2}$ with decreasing momentum $q$ is explicitly represented by the power-law decaying tail of DOS on either (both) spectral edge(s) in the localized (extended) phase, see the inset to Fig. 2(a).

The finite-size mobility edge $E^* \simeq V_{\mathbf{q}_*}$ found numerically can be determined by Ioffe-Regel criterion. Indeed, a state is localized as soon as its localization length $\ell_{loc}$ is smaller than the system dimension $L$. In 2d systems the localization length is exponentially growing with the mean-free path $\ell_{loc} \sim e^{ck_F \ell_{mfp}}$, with a certain constant $c \sim O(1)$. Fixing the Fermi momentum at $k_F = q$ one calculates the mean-free path $\ell_{mfp}(q) \simeq v_q \tau_q$ via the group velocity $v_q = dV_{\mathbf{q}}/dq \sim q^{a-d-1}$ at the momentum $q$ and the level broadening determined by the Fermi Golden rule $\Gamma = \tau_q^{-1} \sim W^2 \rho(E_{\mathbf{q}}) \sim W^2 q^{2d-a}$ for the plane wave scattering on impurities $\mu_i \simeq W$. This gives the fraction $f_{\mathrm{erg}}$ of ergodic extended states below the finite-size mobility edge, $\ell_{loc} \sim L$,

$$f_{\mathrm{erg}} = \pi q_*^2 \sim \left[ W^2 \ln L \right]^{-1/(3-a)} , \tag{23}$$

which decays only as a power of the logarithm of the system size, see Fig. 9.

In the next parts of this section we provide the sketch of the renormalization group approach to the localization at $a < d$, see Sec. 5.1, and the Ioffe-Regel argumentation for the location of the finite-size mobility edge and the corresponding fraction of the ergodic states in this localized phase, see Sec. 5.2.

## 5.1 Main idea of the renormalization group analysis

In this Section we follow [17, 18, 47] and reproduce the idea of the renormalization group (RG) analysis for the 2d anisotropic system. The main assumption of this RG written in the limit of large disorder strength $W \gg 1$ is that at each step, as we consider only the hopping terms at the distance $R$, the localization length $\ell^{(R)}$ of an eigenstate $\left| \psi_n^R \right\rangle = \sum_i \psi_n^R(i) \left| i \right\rangle$ around its maximum $i = n$ is small compared to $R \gg \ell^{(R)}$.

For clarity let's consider the first step of the RG. Similarly to [18] we take the disorder amplitude $W \gg 1$ to be large compared to the nearest-neighbor hopping $V_{i,i+1}$ and apply the

RG procedure to study this problem. As a step of the RG we first cut off the tunneling at a certain scale $R_0$ and calculate the wave functions ($R_0$ modes) for this scale. Then new cutoff $R_1 \gg R_0$ is chosen and new modes ($R_1$ modes) are constructed as a superposition of $R_0$ modes. The localization length increases from $\ell_0 \lesssim R_0$ to $\ell_1 \lesssim R_1$ due to the presence of resonances. At large disorder strength $W \gg 1$ only pairs of resonances are relevant, as the probability to realize triple or higher order ones is smaller in $1/W$ (please see [17] for more details).

The projectors $|\psi_k^{(1)}\rangle$ on new $R_1$ modes can be written via the initial site projectors $|i\rangle$ as follows

$$|\psi_k^{(1)}\rangle = \sum_i \psi_k^{(1)}(i)|i\rangle. \tag{24}$$

Thus, the hopping term $V_{ij} = -\frac{1-\beta \cos^2 \phi_{ij}}{r_{ij}^a}$ rewritten in new operators takes the form

$$\sum_{i,j} V_{ij}|i\rangle\langle j| = \sum_{k,l} |\psi_k^{(1)}\rangle\langle\psi_l^{(1)}| \sum_{i,j} \psi_k^{(1)}(i)\psi_l^{(1)*}(j)V_{ij}. \tag{25}$$

According to RG assumption the modes $\psi_k^{(1)}(m)$ are localized $r_{km} < \ell_1$ at the length, much smaller than the distance to the next resonance $\ell_1 \lesssim R_1$, thus, one can neglect the difference between $V_{ij}$ and $V_{kl}$ $\left(|r_{ij} - r_{kl}| < r_{ik} + r_{jl} < 2l_1 \lesssim R_1\right)$. As a result, Eq. (25) reads as

$$\sum_{i,j} \frac{1-\beta \cos^2 \phi_{ij}}{r_{ij}^a}|i\rangle\langle j| \simeq \sum_{n,m} l_m l_n \frac{1-\beta \cos^2 \phi_{mn}}{r_{mn}^a}|\psi_m^R\rangle\langle\psi_n^R|, \tag{26}$$

with the effective charge $l_n = \sum_i \psi_n^R(i)$.

In order to estimate the renormalized hopping term $l_k l_l^*/r_{kl}^a$ we consider the mean squared value of $l_k$ at a certain energy $E$ as follows [47]

$$\langle l^2\rangle_E = \frac{\langle \sum_k l_k^2 \delta(E-E_k)\rangle}{\rho(E)} = \frac{\left\langle \sum_k \sum_{|i-k|<R_1}^{i,} \sum_{|j-k|<R_1}^{j,} \psi_k^{(1)}(i)\psi_k^{(1)*}(j)\delta(E-E_k)\right\rangle}{\rho(E)}$$
$$\simeq \frac{\sum_{|\mathbf{i}-\mathbf{j}|<R_1} \langle \text{Im } G_{\mathbf{i}-\mathbf{j}}\rangle}{\pi\rho(E)} \simeq \frac{\text{Im } \bar{G}_{q\simeq 1/R_1}(E)}{\rho(E)}. \tag{27}$$

Here the result is written in terms of the density of states (DOS)

$$\rho(E) = \left\langle \sum_k \delta(E-E_k)\right\rangle = \frac{1}{L^d} \sum_{\mathbf{q}} \text{Im } \bar{G}_{\mathbf{q}}(E), \tag{28}$$

and the Green's function

$$G(E+i\eta) \equiv \frac{1}{E+i\eta - H}, \tag{29}$$

averaged over the diagonal disorder $\bar{G}(E) = \langle G(E+i\eta)\rangle$ and, thus, diagonal in the momentum space. The latter reads as

$$\bar{G}_{\mathbf{q}}(E) = \frac{1}{E - \Sigma - V_{\mathbf{q}}}, \tag{30}$$

with the self-energy given by a simplest coherent potential approximation $\Sigma = -\frac{W^2}{12}\bar{G}_0(E)$, consistent with the Fermi Golden rule result

$$\Gamma \equiv \text{Im } \Sigma = -\rho(E)\frac{W^2}{12}. \tag{31}$$

Here we consider for simplicity the box distribution of the disorder $-W/2 < \mu_i < W/2$ with the finite variance $\langle \mu_i^2 \rangle = W^2/12$ and use it in the determination of the self-energy of the Green's function.

In the coherent potential approximation, for the bulk of the spectrum $E \sim W$, DOS is $q$-independent and is determined solely by the disorder amplitude (like in [17]), $\rho(E) \sim 1/W$. Thus, the imaginary part of the Green's function is given by a Lorenzian

$$\text{Im}\,\bar{G}_q(E) \simeq \frac{W}{(E-V_\mathbf{q})^2 + \pi W^2/12} \,, \tag{32}$$

and, thus, the mean squared charge (27) takes the form of the integral over the momentum orientation angle $\phi_\mathbf{q}$, with $q \simeq 1/R \ll 1$

$$\langle l^2 \rangle_E = \frac{W}{2\pi^2} \text{Im} \int_0^{2\pi} \frac{d\phi_\mathbf{q}}{E - V_\mathbf{q} - \Sigma} = \frac{W}{2\pi^2} \int_0^{2\pi} \frac{\Gamma d\phi_\mathbf{q}}{(E - V_\mathbf{q} - \text{Re}\,\Sigma)^2 + \Gamma^2} \,. \tag{33}$$

At $a < d$ the integrand denominator dominated by the hopping spectrum, $V_\mathbf{q}$ has infrared divergence, so the angle averaging depends on whether $V_\mathbf{q}$ versus $\phi_\mathbf{q}$ changes the sign or not for $q \simeq 1/R \ll 1$.

Indeed, for sign-definite $V_\mathbf{q}$, Eq. (22), the above integral is given mostly by $\langle l^2 \rangle_E \sim W\Gamma/V^2_{\mathbf{q} \simeq 1/R} \sim W^2 R^{2(a-d)}$ and leads to

$$\text{Im}\,\bar{G}_{q \sim 1/R_1}(E) \simeq \frac{W}{R_1^{2(d-a)}} \,, \tag{34}$$

and the effective hopping for all $a$ within the RG approximation scales as

$$V_R^{eff} = \min\left( \frac{1}{R^a}, \frac{W^2}{R^{2d-a}} \right) \,, \tag{35}$$

giving localization with the characteristic change of the power law tail at $R \simeq W^{1/(d-a)} \gg 1$. This result can be equivalently obtained from the matrix-inversion trick [16]. More rigorous calculations done at $a = d = 2$ [17] give logarithmic corrections leading to (15).

In the opposite case of $a < a_{AT}$, when $V_\mathbf{q}$ changes sign with respect to $\phi_\mathbf{q}$, simple calculations give $\langle l^2 \rangle_E \sim W/V_{q \simeq 1/R} \sim R^{a-d}$ resulting in $|\psi_E(i)|^2 \sim r^{-d}$. This critical behavior, formally equivalent to the critical case of $a = d$ for the random-sign hopping term $h_{ij}/r_{ij}^a$, hints that the delocalized phase at $a < a_{AT}$ is nonergodic. However, more rigorous calculations of transport based on Kubo formula [35] give logarithmic corrections leading to ergodic behavior.

This analysis puts the basis under the simple localization-delocalization arguments given by Eqs. (21) and (22) and Fig. 11 about the presence of high-energy states on either or both spectral edges.

## 5.2 Ioffe-Regel criterion for the fraction of high-energy ergodic states

Here we estimate the energy-dependent mean-free path for $a < d$ and, based on the Ioffe-Regel criterion of localization, estimate the location of the finite-size mobility edge as well as the fraction of ergodic states in the localized phase of the considered anisotropic model, Eq. (1).

The mean-free path at a certain energy $E$ can be estimated as follows

$$\ell_{mfp}(E) \simeq v_{q_E} \tau_{q_E} \,, \tag{36}$$

where the energy-dependent momentum $q_E$ and the corresponding group velocity $v_q$ are determined from the following equations

$$E = V_{q_E \lesssim 1} \sim q^{a-d}, \Rightarrow q_E \sim \min\left[1, E^{-1/(d-a)}\right], \tag{37}$$

$$v_q = \frac{dV_{\mathbf{q}}}{dq} \sim q^{a-d-1}, \tag{38}$$

while the level broadening $\Gamma \equiv \tau_{q_E}^{-1}$ can be estimated with Fermi Golden rule of the scattering of plane waves on the impurities $\mu_i \sim W$ (see Eq. (31))

$$\tau_{q_E}^{-1} = \operatorname{Im} G_{i-j=0}(E) \simeq \rho(E) \frac{W^2}{12}. \tag{39}$$

Small $q_E$ corresponds to large energies $E \gg W$, thus, the DOS at such energies is not determined by $\rho(E) \sim 1/W$, but involves $q_E$ as follows

$$\rho(E \gg W) = \frac{d^d q_E}{dV_{q_E}} \sim q_E^{2d-a}. \tag{40}$$

As a result, we obtain

$$\ell_{mfp}(E) \sim W^{-2} q_E^{2a-3d-1} \sim W^{-2} E^{(3d+1-2a)/(d-a)}. \tag{41}$$

According to the Ioffe-Regel criterion the states are delocalized

- in $d = 1$ as soon as $\ell_{loc} \sim \ell_{mfp} > L$;

- in $d = 2$ as soon as $\ell_{loc} \sim e^{cq_E \ell_{mfp}} > L$;

- in $d = 3$ as soon as $q_E \ell_{mfp} > 1$.

leading to a certain upper cutoff $q_E < q_*$. The fraction of such delocalized states is given by

$$f_{\text{erg}} = \int_0^{q_*} d^d q \sim q_*^d. \tag{42}$$

After straightforward algebra the mobility edge can be estimated as

- in $d = 1$
$$q_E < q_* = \left(W^2 L^d\right)^{-\frac{1}{2(2-a)}} \Rightarrow f_{\text{erg}} \sim q_* \sim L^{-\frac{d}{2(2-a)}}; \tag{43}$$

- in $d = 2$
$$q_E < q_* = \left(W^2 \ln L^d\right)^{-\frac{1}{2(3-a)}} \Rightarrow f_{\text{erg}} \sim q_*^2 \sim \ln L^{-\frac{d}{3-a}}; \tag{44}$$

- in $d = 3$
$$q_E < q_* = W^{-\frac{2}{9-2a}} \Rightarrow f_{\text{erg}} \sim q_*^3 \sim O(1). \tag{45}$$

In the 2d case of the considered model (see Figs. 9 and 10) the fraction of ergodic states decays as the power of the logarithm of $L^d$. This Ioffe-Regel-based estimate coincides with the fraction of states which are non-ergodic or localized in the momentum $q$-basis, see [48].

If instead, following [16, 49], one calculates the fraction of modes which are localized in the momentum $q$-basis, this condition will be more restrictive for all $d > 1$ as it provides the fraction of plane-wave states, while most of delocalized modes in $d \geq 2$ are of diffusive nature. Such a condition is related to the level spacing $|V_{q_E} - V_{q_E+\pi/L}|$ to be of the order of the corresponding hopping

$$|V_{q_E} - V_{q_E+\pi/L}| \sim \frac{v_{q_E}}{L} > \frac{W}{L^{d/2}}, \tag{46}$$

which leads to

- in $d = 1$

$$q_E < q^{**} = \left(\frac{L^{d/2}W}{t_0}\right)^{-\frac{1}{(2-a)}} \simeq q_*, \tag{47}$$

- in $d = 2$

$$q_E < q^{**} = \left(\frac{W}{t_0}\right)^{-\frac{1}{3-a}} \ll q_*, \tag{48}$$

- in $d = 3$

$$q_E < q^{**} = \left(\frac{W}{L^{1/3}t_0}\right)^{-\frac{1}{4-a}} \ll q_*. \tag{49}$$

## 6 Conclusion and Outlook

To sum up, we explicitly show both numerically and analytically the phenomenon of the anisotropy-mediated reentrant Anderson localization transition relevant for generic 2d quantum dipole models. The transition is demonstrated to occur at a finite anisotropy-tilt angle of dipoles depending on the exponent $a$ of the generalized dipole-dipole interaction controlling excitation hopping. Moreover, close to the pure 2d dipole-dipole interaction $1 < a \leq 2$ the phase diagram has a reentrant nature showing the localization both at large and small tilts, given by the rotation symmetry of the system accompanied by the tilt transformation.

The further research should take into account the robust localized nature of eigenstates in dilute dipolar systems with respect to the ones with randomized interaction strength. This difference between the models with deterministic and random interactions plays an important role also in dense systems where the many-body localization has different properties for such systems (see, e.g. [50–62]). It might be interesting to understand whether there is a many-body localization transition driven by the anisotropy of long-range couplings in such systems.

## Acknowledgements

We thank V. E. Kravtsov for illuminating discussions. We also thank G. V. Shlyapnikov and Luis Santos for previous collaboration on related topics. This research was supported by Russian Science Foundation, Grant No. 21-12-00409 (I. M. K.), by Carrol Lavin Bernick Foundation Research Grant (2020-2021) and by NSF CHE-2201027 grant (A. L. B.) by the DFG projects SA 1031/11, SFB 1227 DQ-mat, by the Federal Ministry of Education and Research of Germany (BMBF) in the framework of DAQC. (X. D.), and by the European Research Council under the European Unions Seventh Framework Program Synergy HERO SYG-18 810451.

**Funding information** This research was supported by Russian Science Foundation, Grant No. 21-12-00409 (I. M. K.), by Carrol Lavin Bernick Foundation Research Grant (2020-2021) and by NSF CHE-2201027 grant (A. L. B.) by the DFG projects SA 1031/11, SFB 1227 DQ-mat, and by the Federal Ministry of Education and Research of Germany (BMBF) in the framework of DAQC. (X. D.).

## A Extrapolation of the fractal dimension in the extended phase

In this Appendix we focus on the extended phase of the considered anisotropic model.

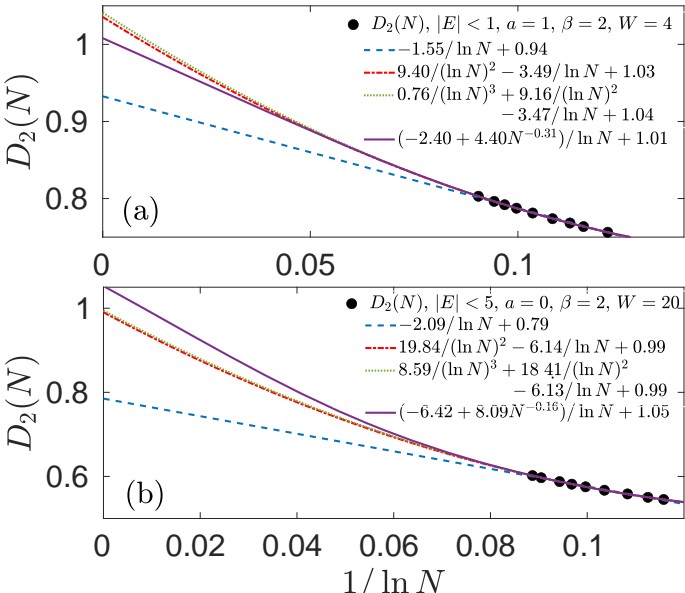

Figure 12: **Finite-size extrapolation of the fractal dimension** $D_2$ (symbols) with linear (blue dashed), quadratic (red dash-dotted), cubic (green dotted) expressions in $x = 1/\ln L^d$ as well as the one with irrelevant exponent (violet solid) considered in [34]. We show two parameter sets (upper panel) $a = 1$, $\beta = 2$ is $W = 4$ and (lower panel) $a = 0$, $\beta = 2$ is $W = 20$ in order to emphasize that this issue present both at weak and strong disorder. $D_2$ is averaged over the energy interval $|E| < W/4$ and extrapolated from $L = 75, 85, 100, 125, 150, 175, 200$, and 250 with the corresponding number of disorder realizations 2000, 2000, 2000, 2000, 1000, 600, 600, and 300, respectively.

First, we should mention that the extrapolation of $D_2$ in this case is more subtle. Due to limited system sizes in 2d the linear approximation (9) provides unreasonable results and, thus, following recent literature some of the authors of this paper use quadratic in $1/\ln L^d$ extrapolation and compare it with further cubic one both for weak and strong disorder, see Fig. 12.[6] In order to double check we also fit the data with the expression with irrelevant exponent suggested in [34]

$$D_q(L) = D_q + \frac{(1-q)^{-1} \ln c_q + \gamma_q L^{-y_{irr}}}{\ln L^d} . \tag{A.1}$$

All the results confirm the ergodic nature of the extended phase in the considered model[7] which is spoiled by severe finite-size effects forcing one to go beyond linear extrapolation, Eq. (9).

---

[6]Here there is an open question whether severe finite-size effects in an ergodic phase are related to weak ergodicity. In this weak ergodic phase the fractal dimensions $D_q \to 1$, but along the path different from the ones from the random-matrix theory, due to the occupation of only a finite fraction of the total Hilbert space by eigenstates mediated by the breakdown of the basis-rotation invariance [63–65]. This phase plays an important role in several recent papers [16, 26, 39, 66–68].

[7]Unlike the long-range static [69–71] and short-range driven [72] models with correlated (quasiperiodic) on-site disorder, in the current model multifractality does not emerge in the extended phase due to the mixture of localized and ergodic states.

# B  Wavefunction spatial decay

Similar to Figs. 6(b) and 7(b) in the main text and the results of [15–17], we consider the typical wave function spatial decay with the distance with respect to its maximum. Fig. 13 confirms the duality of power-law spatial decay rate $\gamma(a) \approx \gamma(2d-a)$ [15–17] in the localized phase of the anisotropic model between the standard locator expansion states $a > d = 2$ and beyond it $a < d$.

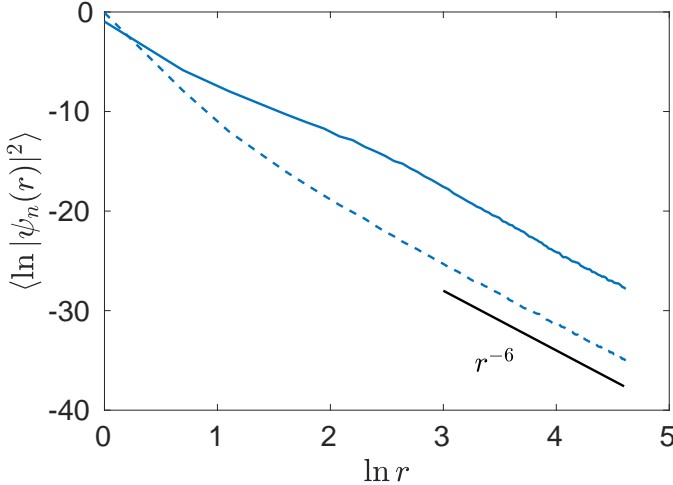

Figure 13: **Power-law spatial decay of eigenstates** in the bulk of the spectrum for $a = 1$ (solid), 3 (dashed), $\beta = 0.5$, at the system size $L = 200$ with 200 disorder realizations. The disorder amplitude is taken to be $W = 20$ for $a = 1$ and $W = 200$ for $a = 3$ in order to make the power-law tail dominant on moderate sizes in both cases.

# C  Spectrum of hopping, Eq. (20)

The spectrum of the hopping term $V_{ij} = -\frac{1-\beta \cos^2 \phi_{ij}}{r_{ij}^a}$ from Eq. (1) is given by its Fourier transform due to translation-invariance of hopping

$$V_{\mathbf{q}} = -\sum_{i,j} e^{iq_x(i_x-j_x)+iq_y(i_y-j_y)} \frac{1-\beta \cos^2 \phi_{ij}}{r_{ij}^a}. \tag{C.2}$$

For $a \neq d = 2$ the latter can be calculated in the continuous approximation as

$$V_{\mathbf{q}} = -\int_0^\infty r \, dr \int_0^{2\pi} d\phi \, e^{iqr \cos(\phi-\phi_q)} \frac{1-\beta \cos^2 \phi}{r^a} = c_a q^{a-2} \left[\beta - a - (2-a)\beta \cos^2 \phi_q\right]. \tag{C.3}$$

Here $c_a = \pi 2^{1-a} \frac{-\Gamma(-a/2)}{\Gamma(a/2)}$, $\Gamma(a)$ is a Gamma-function, and $q = \pi n/L$ is the quantized momentum, with integer $n \lesssim L/a_0$, $a_0$ is the inter-atomic distance which we choose to be unity $a_0 \equiv 1$ without loss of generality. The special case of $a = d = 2$ should be considered separately as the result depends explicitly on $a_0$

$$V_{\mathbf{q}} = \pi \left[(2-\beta)(\gamma_E + \ln(qa_0/2)) - \frac{\beta}{2}\cos(2\phi_q)\right], \tag{C.4}$$

with the Euler – Mascheroni constant $\gamma_E \simeq 0.577216$.

The divergence of both Eqs. (C.3) and (C.4) at $q \to 0$ at $a \leq d$ signals on the presence of (the measure zero of) high-energy delocalized states [16, 17].

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
