# Peer review of "Anisotropy-mediated reentrant localization"

_SciPost Physics, doi:SciPost Phys. 13, 116 (2022)_

## Round 1 · Referee Report · Anonymous (Referee 1) · 2022-9-1

Report

The authors corrected the manuscript following earlier remarks. The manuscript may be accepted for publication in its present form

---

## Round 1 · Referee Report · Anonymous (Referee 2) · 2022-10-9

Report

Authors have addressed my concerns from the previous round, I recommend publication.

---

## Round 1 · Author Response

Dear Editor,

Thank you for providing us the second report of the Referee 1. Hereby we resubmit our manuscript after minor corrections, suggested by the Referee 1.

Please see below point-by-point changes in the manuscript and the reply to the Referee. The changes are also marked in the manuscript by the red font and yellow highlights.

Sincerely yours, Xiaolong Deng, Alexander Burin, and Ivan Khaymovich

Reply to the second report of the Referee 1

We are very grateful to the Referee 1 for his/her detailed second review of our manuscript. Below we present the list of amendments resulting from the comments of the Referee.

Weaknesses - Some numerically based conclusions are not convincing.

Reply: We believe that in the revised version of the manuscript the numerically-based conclusions, consistent with the analytical predictions, are convincing for the Referee 1.

Report: I believe the manuscript has been significantly improved with respect to the previous version. It contains significant new results on the rich 2D model, some of them not obvious at first glance. The paper combines pure numerical studies with renormalization group analysis. I hope that the authors may take into account the suggested changes as listed below.

Reply: We thank the Referee 1 for high evaluation of our revised version and provide the reply below.

Requested changes 1. Considering the gap ratio statistics the authors rejected my suggestion to make the more detailed energy dependent study arguing that the energy dependence shows only close to $\beta = a$ transition. While this is not documented in the figures, it is clearly stated in p.8 so I withdraw my suggestion.

Reply: We thank the referee for this.

Requested changes 1. (continuation) Still considering gap ratio statistics Fig.3 shows that finite size scaling of two crossings present in case (c) and (d) yields different $\beta_{AT}$ values. On the other hand, as mentioned in p.5 there is a symmetry relating $0<\beta<2$ interval with $\beta>2$. Is it not this symmetry which leads to the second crossing in panels (c) and (d)? Can one be more quantitative about the relation between the corresponding $\beta_{AT}$ values (1.45 versus 3.1 for panels g,h) ? May be we learn more about the consistency of the finite size scaling from such a comparison?

Reply: Yes, indeed, the pairs of crossings in panels (c) and (d) of Fig. 3 are related to each other by the exact symmetry (3) of the model: the fact that the critical exponents are the same in each of the pairs implicitly confirms this. In the revised version, in order to make the symmetry clear, we have added the clarifying phrase into the text: “Note that the pairs of crossings $\beta_{AT}$ in Fig. 3(c, d) are related to each other with respect to the symmetry (3) within the above mentioned error bar, while the critical exponents are just the same.”. In addition, we have added the error bars for $\beta_{AT}$ given by $\pm 0.05$ for $\beta<2$ and by $\pm 0.1$ for $\beta>2$.

Requested changes 2. This referee is entirely lost with Fig.4. The caption and the horizontal axis says $D_2$ versus $a$. On the other hand the points both yellow and blue are denoted by $a=0$. Is it not a bit inconsistent? Caption says yellow squares are for $W=10$ while in the figure $W=20$ is indicated. Something is simply wrong here.

Reply: We thank the Referee 1 for pointing out these typos. In the revised version of the manuscript we have modified both the figure and the caption accordingly.

Requested changes: 3. Fig.5 seems to show that three color lines for different values of L yield the extrapolated black curve which is far far above finite size results. Is such an extrapolation not too courageous? What are the errors of this procedure?

Reply: Yes, indeed, Fig. 5 shows the extrapolation of the finite size spectrum of fractal dimensions $f(\alpha, L)$ to the infinite system size. As usual (see, e.g., [16, 17, 37]) with $f(\alpha)$ the main finite size correction of $f(\alpha, L)$ is given by the vertical shift (i.e. the weakly $L$-dependent prefactor in the probability distribution of $\alpha$, given by subleading corrections in (12)). As the shape of $f(\alpha, L)$ is practically the same for all the system sizes that we considered, we just get rid of the (unknown) prefactor by extrapolation (12). The deviations of the extrapolated value of the maximum of $f(\alpha)$ from $1$ gives an estimate for the error bar of this procedure.

Requested changes: 4. The authors are asked to review and correct some formulae. In (17) the equality with Dirac delta function is of course wrong for $t>0$, also imaginary unit "i" is missing from exponents in (17) and (18).

5. Few places require editing. Eg. p. 3 "By tailor the optical forces" should be "Tailoring the optical forces"; Double dot ending para after (2) in p. 5 is not needed etc…

Reply: We thank the Referee 1 for pointing out the misprints and correct them accordingly in the revised version.

---

## Round 1 · List of Changes

1. Page 3: the phrase "By tailor the optical..." has been changed to "Tayloring the optical...".
  2. Page 5: the footnote, clarifying the definition of the fractal dimension has been added.
  3. Page 7: the error bar for $\beta_{AT}$ has been added after Eq. (7).
  4. Page 8: the symmetry (3) has been discussed with respect to two crossing points in Fig. 3(c, d) and the coincidence of the critical exponents $\nu$ has been emphasized.
  5. Page 9, Fig. 4: the labels and the caption have been corrected and clarified.
  6. Page 12: the typos in Eqs. (17-18) have been corrected.
  7. Page20: the grant number in the Acknowledgement section has been updated.

---

## Editorial Decision

published